# Characterization of N-Terminal Asparagine Deamidation and Clipping of a Monoclonal Antibody

**DOI:** 10.3390/antib12030059

**Published:** 2023-09-19

**Authors:** Jing Zhen, Jennifer Lee, Yueyang Wang, Lena McLaughlin, Fei Yang, Zhengjian Li, Jihong Wang

**Affiliations:** Department of Analytical Sciences, U.S. Technical & Biologics Development, Horizon Therapeutics, Rockville, MD 20850, USA

**Keywords:** monoclonal antibody(s), biopharmaceutical characterization, deamidation, N-terminal truncation, charge profile, liquid chromatography–mass spectrometry (LC-MS)

## Abstract

This study presents a novel degradation pathway of a human immunoglobulin G (IgG) molecule featuring a light chain N-terminal asparagine. We thoroughly characterize this pathway and investigate its charge profiles using cation exchange chromatography (CEX) and capillary isoelectric focusing (cIEF). Beyond the well-documented asparagine deamidation into isoaspartic acid, aspartic acid, and succinimide intermediate, a previously unreported clipping degradation pathway is uncovered. This newly identified clipped N-terminal IgG variant exhibits a delayed elution in CEX, categorized as a “basic variant”, while retaining the same main peak isoelectric point (pI) in cIEF. The influence of temperature and pH on N-terminal asparagine stability is assessed across various stressed conditions. A notable correlation between deamidation percentage and clipped products is established, suggesting a potential hydrolytic chemical reaction underlying the clipping process. Furthermore, the impact of N-terminal asparagine modifications on potency is evaluated through ELISA binding assays, revealing minimal effects on binding affinity. Sequence alignment reveals homology to a human IgG with the germline gene from Immunoglobulin Lambda Variable 6-57 (IGLV6-57), which has implications for amyloid light-chain (AL) amyloidosis. This discovery of the N-terminal clipping degradation pathway contributes to our understanding of immunoglobulin light chain misfolding and amyloid fibril deposition under physiological conditions.

## 1. Introduction

In biologics development, it is essential to understand various post-translational modifications (PTMs) and their potential impacts on clinical outcomes such as pharmacokinetics and efficacy. The most common PTMs for monoclonal antibodies (mAbs) include deamidation, isomerization, N-terminal pyroglutamate (pyroE) formation, C-terminal lysine cleavage and C-terminal amidation [1,2,3,4,5,6,7,8,9,10,11,12]. It has been reported that deamidation and isomerization, especially within the antibody complementarity-determining regions (CDRs), lead to reduced potency [1,3] in the Fc region that can abolish effector function [4]. Antibodies with high mannose glycans in the antibody Fc region have a shorter half-life compared with other glycan structures [5]. Therefore, it is a critical part of product quality control and a regulatory requirement to characterize PTMs in antibody therapeutics to assess if they are critical quality attributes (CQAs) [13]. Asparagine deamidation, a crucial contributor to charge variant, occurs when the backbone nitrogen engages in a nucleophilic attack on the side chain carbonyl carbon, and subsequently loses an ammonia, resulting in the formation of a succinimide intermediate. The succinimide intermediate product can be hydrolyzed further into isoaspartic acid (isoD) and aspartic acid (D) [11]. Deamidation of recombinant mAbs and endogenous immunoglobulin G (IgG) antibodies can occur *in vivo* in both humans and monkeys [14]. Deamidation to iso-D and D reduces protein isoelectric point (pI) values, leading to an increased percentage of acidic peak variants in cation exchange chromatography (CEX) chromatograms and capillary isoelectric focusing (cIEF) electropherograms. CEX utilizes a negatively charged stationary phase to separate protein molecules by different charge profiles. The more positively charged proteins will bind to the stationary phase and is a later elution on the chromatogram. cIEF utilizes a pH gradient and electric field to migrate proteins with different charges. Each protein will migrate to the position where the pH equals its pI and the protein has a net charge of zero. Another commonly observed modification is N-terminal pyroglutamate formation [15], in which either terminal glutamic acid (E) or glutamine (Q) cyclizes into pyroE with the loss of either water or ammonium. These PTMs can cause shifting of pI values, where variants with lower pI values than the main peak are defined as “acidic”, and variants with higher pI values are defined as “basic”. CEX and cIEF are two commonly used tools for biologic charge variants analysis [16]. In most situations, CEX chromatograms are similar to cIEF electropherograms [17]. Another one of the most used tools to characterize these PTMs is via enzymatic digested peptide mapping followed by liquid chromatography (LC) and mass spectrometry (MS) analysis [18]. Proteinases, such as trypsin, digest protein molecules into smaller peptides and this is followed by peptide separation by reversed-phase chromatography and identification by mass spectrometry. During biologic therapeutics development, a forced degradation study is conducted to enrich the degradation PTMs for characterization and support critical quality attributes (CQA) determination.

In this study, a unique therapeutic IgG molecule with N-terminal asparagine showed an unusually increased percentage rate of the acidic peak measured by cIEF under both accelerated and stressed conditions. Further characterization of the IgG was performed to understand the potential root causes contributing to the acidic peak formation. All potential N-terminal asparagine deamidated products including iso-D, D, and succinimide intermediates were observed in this study, and their distributions in CEX and cIEF were comprehensively analyzed and compared. Interestingly, we also observed that the charge profiles generated by CEX and cIEF were different with heat-stressed samples, and a unique basic peak variant was detected by CEX. Further investigation through the fractionation of CEX basic peaks and in-depth characterization by peptide mapping and intact/reduced masses indicate the enrichment of an uncommon N-terminal asparagine clipping on the light chain of this IgG. Further characterization of stressed samples suggests that N-terminal asparagine clipping occurs with increased temperature and incubation time, which correlates with its deamidation level. Binding capacity determined by enzyme-linked immunosorbent assay (ELISA) suggests this deamidation/clipping has limited impact on the binding to its therapeutic target. This is the first study to characterize N-terminal asparagine deamidation, its clipping, and its impact on mAb CEX and cIEF profiles, which are part of routine release testing methods in biologics manufacturing. This study not only comprehensively investigated the underlying mechanism of this IgG molecules’ degradation, but it also provides insights into other similar antibody molecules’ characterizations and quality control for the future.

## 2. Materials and Methods

### 2.1. Materials

mAb-A was a purified monoclonal antibody Drug Substance produced in house and formulated at pH 6.0. Trypsin enzymes were obtained from Promega (Madison, WI, USA Cat# V5113). Lys-C was purchased from Fujifilm Wako (Osaka, Japan. Cat# SKR0971). Urea (OmniPur^®^, Product# 29700) and Trizma^®^ hydrochloride solution (Cat# T7943) were obtained from Millipore Sigma (Billerica, MA, USA). LC–MS grade water (Cat# W6-1), acetonitrile (Cat# A955-500), formic acid (Cat# A117-50) and trifluoroacetic acid (Cat# A116-1AMP) were obtained from Fisher Chemical (Fair Lawn, NJ, USA). Bovine Serum Albumin (BSA, Cat# A2058-25G), Sodium phosphate dibasic (Cat# S3264-250G), sodium phosphate monobasic monohydrate (Cat# S5011-500G), and sodium chloride (Cat# S9625-1KG) and guanidine hydrochloride solution (Cat# G7294-100ML), and 4M urea (Cat# 51456-500G) were obtained from Sigma-Aldrich (St. Louis, MO, USA). Iodoacetamide (IAA, Cat# 90034) and dithiothreitol (DTT, Cat# 20291) were obtained from Thermo Fisher Scientific (Waltham, MA, USA). pI markers 7.05 (Cat# 046-032) and 9.99 (Cat# 046-034), 1% methyl cellulose (Cat# 101876), and 500 mM arginine (Cat #042-6910) were obtained from ProteinSimple (San Jose, CA, USA). Pharmalyte 3–10 was obtained from Cytiva (Uppsala, Sweden; Cat# 17045601).

### 2.2. Degraded Samples Preparation

A total of 100 mg/mL of antibody in formulation buffer was diluted to 10 mg/mL by adding 20 mM Histidine, with pH 6.0 and pH 7.4, respectively. Then, four aliquots of 500 µL of each were placed separately in stability chambers of 25 °C and 40 °C for either 2 weeks or 1 month.

### 2.3. IEC Chromatography and Fraction Collection

The experiment was performed on a ProPac Elite WCX, 4 × 150 mm column (Thermo Fisher, Waltham, MA, USA, PN# 302972); Mobile phase A was 20 mM phosphate buffered solution, pH 6.3; mobile phase B was 20 mM phosphate buffered solution, pH 6.3 with 250 mM NaCl. The flow rate was 0.5 mL/min and the column temperature was 35 °C. The injection volume was 10 µL and the UV detector wavelength was 280 nm. The gradient was as follows: 0% B in 0–10 min; 0–60% B in 10–31 min; 60–0% B in 31–34 min; 0% B in 34–35 min. For collected fractions from cation exchange chromatography, they were concentrated to 0.5 mg/mL (for B, C, D) or 1.0 mg/mL (for E, F). The clear cut-off time for each collected peak is listed in the Appendix A.

### 2.4. Imaged Capillary Isoelectric Focusing

Imaged capillary isoelectric focusing was performed on a Maurice Analyzer (ProteinSimple, San Jose, CA, USA). The experiment was conducted by first diluting all fraction samples over 1 mg/mL to 1 mg/mL with water. All samples were further diluted with a mixture of water, pI markers 7.05 and 9.99, pharmalyte 3–10, 1% methyl cellulose, 500 mM arginine, and 4 M urea. System suitability was prepared freshly on the day of analysis. All samples and system suitability were then loaded into a 96-well plate and inserted into the Maurice.

### 2.5. Intact Mass Analysis

A total of 2 µL of the concentrated fractions from the cation exchange chromatography step was injected into LC–MS for intact mass analysis. For deglycosylated intact mass analysis, 20 µL of each sample was added with 20 µL of 100 mM Tris-HCl buffer, pH 7.6 and 1 µL of PNGase F (10 U) and incubated at 37 °C for 4 h. Further reduction was performed by the addition of 1 µL of 500 mM DTT with incubation at 37 °C for 30 min. For LC–MS analysis, mobile phase A was water with 0.02% TFA (trifluoroacetic acid), and mobile phase B was acetonitrile with 0.02% TFA. The column was MAbPac RP; 4 µm 2.1 × 100 mm. The flow rate was 0.2 mL/min and the column temperature was 70 °C. For intact mass analysis, the gradient started from 20% B at 0–0.1 and increased to 35% at 5 min. Then the gradient ramped up to 90% B at 2 min and held until 4 min. Then it decreased back to 20% B at 4.5 min and remained the same until 8 min. For reduced mass analysis, the gradient started from 20% B and held for 0.5 min and further increased to 50% B at 9 min, 90% B at 10.6 min and held until 12.5 min. Then the gradient decreased to 20% B at 13 min and held for another 2 min. For mass spectrometer parameters, the spray voltage was 3.5 kV, with a vaporizer temperature of 275 °C and ion range transfer tube temperature of 300 °C. The scanning mass range was 800–4000 *m*/*z* for intact mass analysis and 600–3000 *m*/*z* for reduced mass analysis with a resolution of 15,000.

### 2.6. Tryptic Peptide Mapping

For trypsin digest, samples were denatured using an 8 M guanidine, pH 7.6 denaturing buffer followed by reduction by DTT and alkylation by IAA. The reduced and alkylated samples were buffer exchanged into a solution containing 2 M urea and 100 mM Tris at pH 7.6 trypsin was then added (enzyme-to-protein 1:12.5) and incubated at 37 °C for 4 h. The digestion was stopped by TFA followed by analysis using a Vanquish Horizon liquid chromatography system with an Exploris 480 orbitrap mass spectrometer (Thermo Fisher, Waltham, MA, USA). Mobile phase A was water with 0.02% TFA and mobile phase B was acetonitrile with 0.02% TFA. The gradient was as follows: 1% B in 0–2 min, 1–5% B in 2–5 min; 5–20% B in 5–29 min; 20–21% B in 29–34.25 min; 21–24% B in 34.25–38.5 min; 24–35% B in 38.5–70 min; 35–90% B in 70–77.5 min; 90% B in 77.5–82.5 min; 90–1% B in 82.5–84.5 min; 1% B in 84.5–86 min; 1–10% in 86–88.5 min; 10–45% B in 88.5–96.5 min; 45–90% B in 96.5–98 min; 90% in 98–104 min; 90–1% in 104–106 min; 1% B in 106–120 min. The column used was an Acclaim RSLC C18 reversed-phase column (2.2 µm, 2.1 × 150 mm). The mass spectrometry parameters were as follows: spray voltage of 3.8 kV; ion transfer tube temperature of 320 °C with a vaporizer temperature of 150 °C; scan range of 300–2000 *m*/*z* with a resolution of 60,000. 

### 2.7. Lys-C Peptide Mapping

Lys-C digest: 25 µL of each sample was mixed with 30 mg of Guanidine-HCl in an Eppendorf tube and 0.5 µL of 500 mM DTT. The mixture was incubated for 30 min at 37 °C, followed by the addition of 1 µL of 500 mM IAA and further incubation in the dark at room temperature for 20 min. Afterward, 5 µL of Lys-C enzyme solution (0.5 mg/mL) and 130 µL of digest buffer (100 mM Tris-HCl, pH 7.6) were added followed by incubation for 2 h at 37 °C. Afterward, another 5 µL Lys-C was added and further incubation for 2 h before LC–MS analysis. Solvent A was water with 0.02% TFA, and solvent B was acetonitrile with 0.02% TFA. The gradient was as follows: 1% B in 0–2 min, 1–5% B in 2–5 min; 5–20% B in 5–29 min; 20–21% B in 29–34.25 min; 21–24% B in 34.25–38.5 min; 24–35% B in 38.5–70 min; 35–90% B in 70–77.5 min; 90% B in 77.5–82.5 min; 90–1% B in 82.5–84.5 min; 1% B in 84.5–86 min; 1–10% in 86–88.5 min; 10–45% B in 88.5–96.5 min; 45–90% B in 96.5–98 min; 90% in 98–104 min; 90–1% in 104–106 min; 1% B in 106–120 min. The mass spectrometry parameters were as follows: spray voltage of 3.8 kV; ion transfer tube temperature of 320 °C with a vaporizer temperature of 150 °C; scan range of 300–2000 *m*/*z* with a resolution of 60,000.

### 2.8. Potency Assay

Investigation of mAb binding to its target was performed by an indirect ELISA. In brief, the specific mAb target was generated by Sino Biological (Beijing, China) and was coated onto a 96-well plate overnight. Wells were blocked with 1% BSA the following day and incubated with the mAb containing the N-terminal light chain sequence in a dose-dependent response for 1 h at room temperature. An anti-human IgG detection antibody conjugated to horse radish peroxidase (Jackson ImmunoResearch, Philadelphia, PA, USA; Cat# 109-035-098) was added to each well for 1 h at room temperature. TMB substrate (Surmodics, Eden Prairie, MN, USA; Cat# TMBW-1000-01) was added and reacted with the peroxide in the presence of the bound peroxidase conjugate, resulting in a colored product. Absorbance was measured at 450 nm minus the background measured at 650 nm, generating a signal that was proportional to the potency of the sample. Each sample was run in triplicate, and the average potency was calculated.

## 3. Results

### 3.1. Charge Variants Separations

The unstressed IgG material was tested using both the CEX and cIEF methods for charge variant analysis, and different charge profiles were observed. CEX analysis showed a relative acidic peak area of 19%, a main peak of 78%, and a basic peak of 2.4%. In comparison, cIEF analysis showed a relative acidic peak area of 29%, a main peak area of 66%, and a basic peak area of 4.4%, indicating that cIEF has better resolution of the acidic variants. The antibody was then stressed at 25 °C and 40 °C, respectively, at pH 6.0 for 2 weeks and 1 month, followed by both CEX and cIEF testing. As shown in Figure 1, treatment at 25 °C minorly increased the acidic peak percentage at 1-month incubation from 19% to 24% by CEX and from 29% to 34% by cIEF analysis. Incubation at 40 °C had a noticeable effect on the peak distributions, leading to an increase in the acidic peak percentage and a decrease in the main peak percentage. After a 2-week and a 1-month incubation, the acidic peak percentage rose to 50% and 56% by CEX and to 59% and 78% by cIEF analysis, respectively. The main peak percentage decreased to 34% and 13% by CEX, and to 37% and 19.6% by cIEF analysis, respectively. This high acidic peak percentage increase was further investigated.

Interestingly, there was an increase in the basic peak percentage from 2.5% (unstressed) to 16.1% (2 weeks, 40 °C) and 30.6% (1 month, 40 °C) on CEX chromatograms, whereas on cIEF electropherograms, the basic peak percentage decreased from 4.4% (unstressed), to 3.8% (2 weeks, 40 °C) and 2.7% (1 month, 40 °C). These distinct differences are summarized in Table 1 and were further investigated. A similar trend of acidic peak increase was also observed in forced degradation at 40 °C, pH 7.4 (Appendix A).

### 3.2. Charge Variants Identification

To further elucidate the identity of each charge variant, the sample stressed for 2 weeks at 40 °C and pH 7.4 was selected for fractionation by CEX. The collected fractions were concentrated using a 10K cut-off filter and subsequently tested by cIEF, peptide mapping, and intact mass analysis. As shown in Figure 2, six different fractions (A-F) from CEX were collected and cross analyzed by cIEF. As expected, the peak orders of CEX and cIEF did not match. The acidic Peak B with a lower retention time (24.1 min) on CEX had a higher pI value (8.2) than the later eluting Peak C (pI 8.1) and D (pI 8.1). The basic Peak F on CEX had the same pI value of 8.3 as the main Peak E on cIEF, indicating that this variant cannot be differentiated from the main peak by isoelectric focusing. Fractions A–F were then digested by Lys-C for peptide mapping analysis on LC–MS to further elucidate their identities. 

The peptide mapping study indicates that Peaks A and B are enriched with N-1 asparagine deamidation on the light chain, Peak C is enriched with N-327 deamidation, and Peak D is enriched with N386/391 deamidation. Based on the peak pattern, Peak A is likely to have N-1 deamidation on both light chains, and Peak B has one N-1 deamidation on one of the light chains. The distributions of these acidic variants in different CEX fractions are summarized in Table 2. Interestingly, the basic Peak F on CEX was identified as N-terminal asparagine (N) clipping in the light chain. This identification was based on MS1 intact mass spectra and its MS2 fragment ions mass spectra. The clipped peptide had a molecular weight of 114 Dalton less than the unmodified peptide. As shown in the MS2 fragment ions spectra in Figure 3, the clipped and unmodified peptide shared the same *y* fragment ions, but different *b* fragment ions, supporting the loss of an N-terminal asparagine. 

The N1-clipped peptide was enriched in the CEX fraction F with a relative abundance of 27%, which was significantly higher than in the main peak fraction E with 6%. The original unmodified peptide had a retention time of 31.38 min, whereas the clipped peptide was eluted earlier with a retention time of 30.29 min, caused by the increased peptide hydrophilicity. Further reduced light chain and the heavy chain intact mass analysis demonstrate that fraction F’s light chain has a molecular weight of 23,312 Da, which is ~115 Da less than the major peak in fraction E (23,427 Da). The mass difference correlates with the loss of an N-terminal asparagine molecular weight of 114 Da, further supporting the conclusion of our peptide mapping analysis. No differences were observed between the reduced heavy chains. Detailed deconvoluted mass spectra are listed in the Appendix A. In addition to the clipping, methionine oxidation occurs at the terminal of light chain with a measured *m*/*z* of 877.9222 for the non-clipped peptide and corresponding *m*/*z* of 830.9032 for the clipped peptide. Two deamidated peptides and their succinimide intermediate were also observed. Variants of terminal peptide H1 are summarized in Table 3. Their extracted ion chromatograms are shown in Figure 3. 

Based on the analytical results discussed above, the peak identities measured by CEX and by cIEF are marked in Figure 4. The first group of acidic peaks depicted in the CEX chromatogram (22–23.8 min) correspond to variants with two deamidation sites (N-1, N327 and N386/391); the second group of acidic peaks (23.8–25.3 min) are variants with one deamidation site in the order of N-1, N327 and N386/39; and the basic peaks (25.8–26.0 min) are variants with one or two N-terminal asparagine clippings. In the cIEF electropherogram, the first acidic group (pI 7.40–8.13) contains variants with multiple deamidation sites; the second acidic group (pI 8.13–8.25) contains a mixture of variants with deamidation N-327, N386/391; and the third acidic group (pI 8.25–8.33) is concentrated with N-1 deamidation. 

### 3.3. Protein Stability and Potency Impact 

All samples at different stressed and accelerated conditions were tested by peptide mapping. The deamidation and N-terminal peptide characterizations are summarized in Figure 5. It is commonly recognized that high temperature and pH conditions can increase deamidation levels [19]. To further assess the impact of the N-terminal clipping on antibody potency, a plate-based ELISA potency method was developed and performed to evaluate binding (herein referred to as relative potency compared to a reference standard mAb) between the antibody to its target. Samples in a stability study were stressed for up to six months at 25 °C and up to three months at 40 °C at a pH of 6.0, followed by cIEF and ELISA potency analysis. The potencies from stressed samples were not affected (ranging from 90% to 110% relative potency) even with a corresponding increase in acidic variants percentage of up to 70% (Figure 6A), thus the N-terminal clipping and terminal peptide methionine oxidation were less likely to impact binding between the CDR and its target. CEX fractions were also tested by the ELISA potency method and showed no significant decrease (ELISA variability is typically 80–120%) in relative potency for fractions with different levels of N-terminal deamidation or clipping (Figure 6B). No difference in aggregation levels was observed for this antibody compared to other studied antibody therapeutics with either lambda or kappa light chains. Considering the location of the asparagine, its biological impact may be similar to N-terminal pyroglutamate reported in other studies [16] and could be generally considered as non-CQA.

## 4. Discussion

In this study, it was discovered that there is a significant change in protein charge profiles shown by CEX and cIEF as a result of the degradation of an N-terminal asparagine. Further characterization study by mass spectrometry revealed that N-asparagine deamidation can lead to N-terminal asparagine clipping. This clipped protein is observed in the CEX charge profile as a basic peak and is not separated by cIEF (main peak). The different charge profiles evident from the two methods might be a result of the change in protein surface charge. Further stability studies indicate that there is a positive correlation between N-terminal amino acid clipping and deamidation, thus we propose a tentative degradation pathway for an IgG with an N-terminal asparagine residue (Figure 7). The N-terminal residue of a peptide or protein can result in a succinimide intermediate from asparagine deamidation which can be hydrolyzed from both sides and thus is clipped off from the peptide chain. One question is whether the clipping may be caused by proteases from host cell proteins. To address this, we incubated the IgG in cell culture medium at 40 °C for two weeks and compared it to the cleavage level of the IgG incubated in PBS. No significant difference in N-terminal asparagine clipping rate was observed, which indicates that proteases are unlikely to cause terminal cleavage (Appendix A). This terminal asparagine clipping has not been reported in other proteins yet. Because this phenomenon occurred in both formulation buffer and in PBS, we believe it is structure-related, and is not related to the formulation conditions. The observed analytical profiles will be beneficial for scientists to monitor PTMs using the CEX and cIEF methods, which would be used for in-process and release testing in biologic manufacturing. In this study, the investigation to determine root cause is preliminary. There are other potential causes of this observed clipping phenomenon which need further confirmatory studies in the future. 

Additional protein sequence BLAST analysis and alignment show that a similar terminal sequence NFMLTQPHSVSESPG comes from the germline gene of a human immunoglobulin lambda variable 6-57 (IGLV6-57), which is linked to AL amyloidosis [20,21,22,23,24,25]. In this situation, the misfolded immunoglobulin light chains form amyloid fibrils that can deposit in organs and tissues. N-terminal asparagine clipping could occur under physiological conditions and result in the exposure of a hydrophobic residue, phenylalanine, which potentially could become the seed of light chain amyloid fibrils formation. 

To evaluate if the N-terminal asparagine clipping could cause light chain aggregate formation in IgG, we compared the aggregation level of this IgG with other IgGs without this specific terminal peptide. The results showed no difference in aggregation formation during stability testing (Appendix A), which indicates that the therapeutic mAb with the N-terminal light chain is folded properly and is unlikely to cause any aggregate amyloid fibrils formed from light chain alone with this specific sequence.

The hypothetical link between the N-terminal asparagine clipping and the formation of excessive light chain amyloidosis requires further investigation. Contradictory to other therapeutic antibodies in production, we found very little free light chain in cell harvest for this molecule. One possibility is that the free light chains could be precipitated out due to aggregates formation and are no longer detectable in the cell culture harvest after processing. Additional studies include stably expressing only the light chain and determining its structural stability in the cell harvest. 

It is worth noting that the formation of N-terminal pyroglutamate has been reported to impact the solubility of Aβ38 and Aβ40 and to induce amyloidogenic properties in amyloid peptides [26,27]. Immunotherapy aimed at targeting pyroglutamate-3 Aβ is currently under development [28]. To further explore the connection between N-terminal asparagine clipping and an amyloidosis-associated gene, we can conduct additional investigations. These investigations may involve peptide mapping on samples formed by AL amyloids to determine the presence of N-terminal deamidation and clipping.

The terminal amino acid is relatively distant from the antibody CDR, thus the minimal impact on its potency is expected and confirmed in this study. No higher-order structural changes have yet been reported for N-terminal modifications, such as pyroglutamine formation and clipping. We do not expect this N-terminal deamidation and clipping to impact the higher-order structure of this molecule. The findings in this study can guide biologics process development to characterize therapeutic antibodies within human germlines containing N-terminal asparagine.

This study demonstrated a unique clipping behavior in antibody therapeutics. The characterization work will support analytical method development in support of manufacturing process development and release/stability specification setting in lot release and shelf-life assignment. Fortunately, this N-terminal deamidation and clipping appear to be significant only under accelerated and stressed conditions. When this IgG was stored under its intended storage conditions, either frozen as a Drug Substance or at 2–8 °C for a Drug Product, minimal changes were observed over a three-year period.

## Figures and Tables

**Figure 1 antibodies-12-00059-f001:**
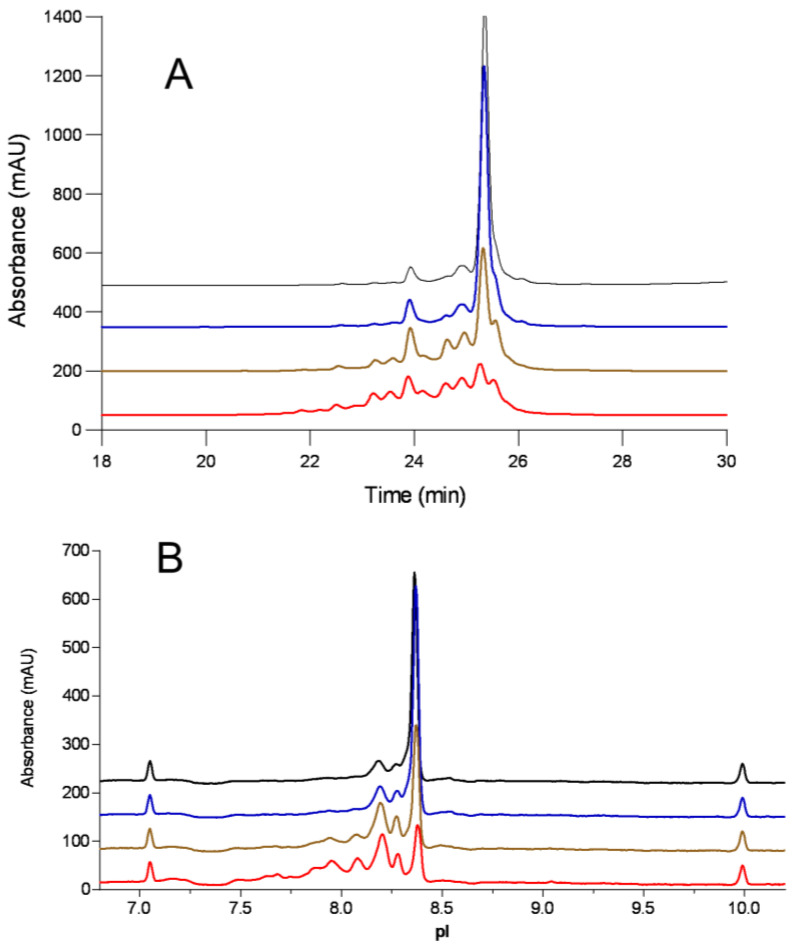
(**A**) Cation exchange chromatogram; (**B**) cIEF electropherogram of samples stressed at pH 6.0 (T0 control (black), 25 °C, 1 month (blue), 40 °C, 2 weeks (brown); 40 °C, 1 month (red)).

**Figure 2 antibodies-12-00059-f002:**
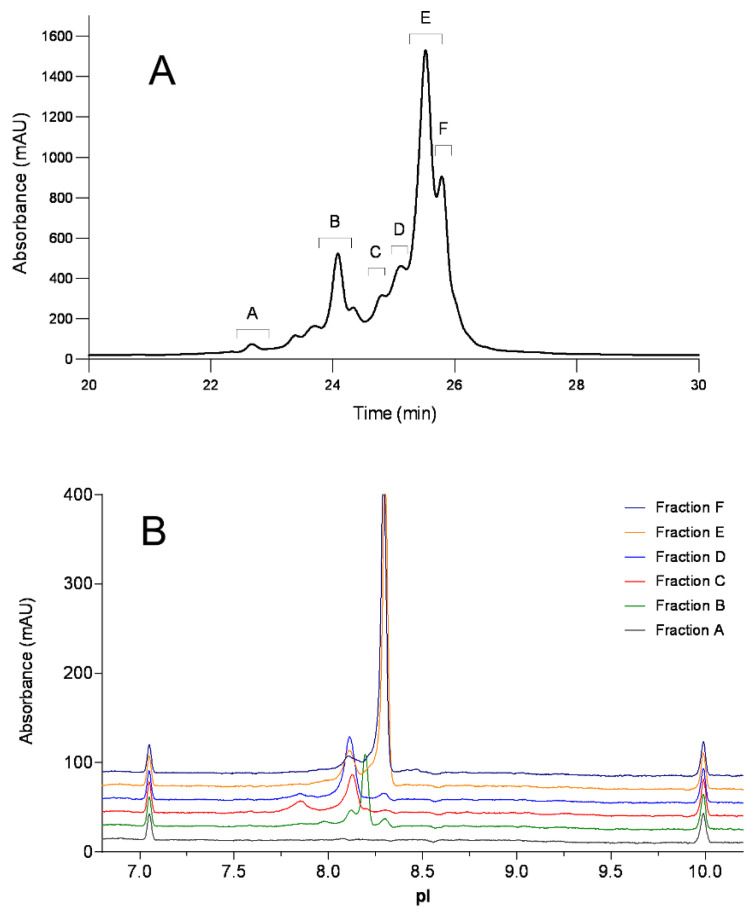
Electropherogram of fractions from CEX (**A**) on cIEF (**B**).

**Figure 3 antibodies-12-00059-f003:**
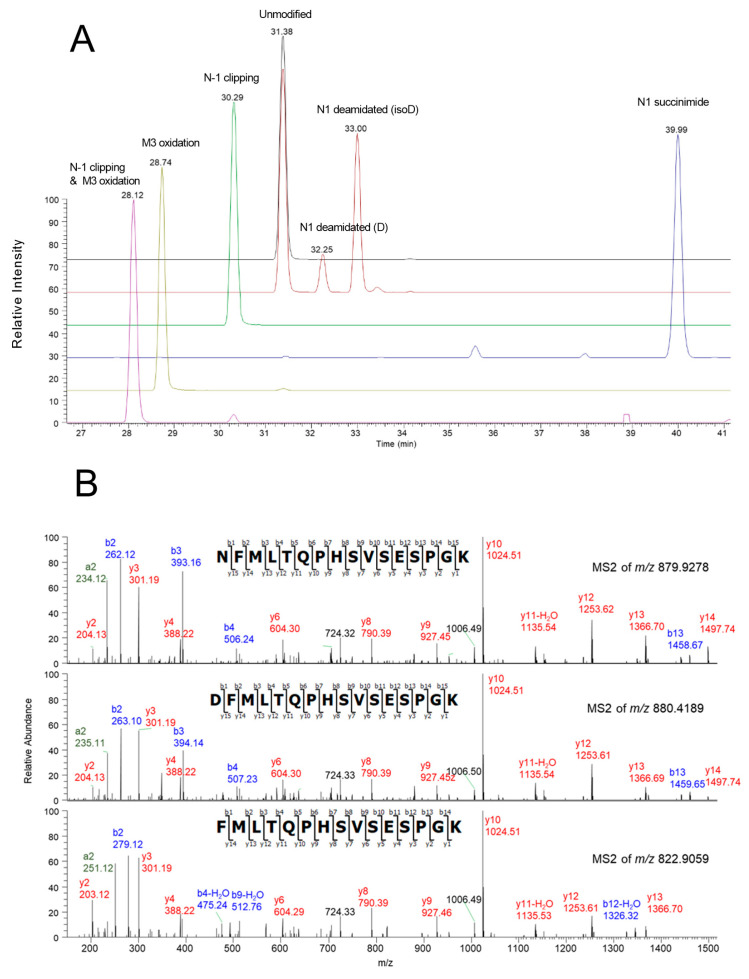
(**A**) Extracted ion chromatograms of peptide L1 and its degradation variants; (**B**) MS2 spectra of peptide L1 (unmodified) and its degradation variants (N-asparagine deamidated and clipped).

**Figure 4 antibodies-12-00059-f004:**
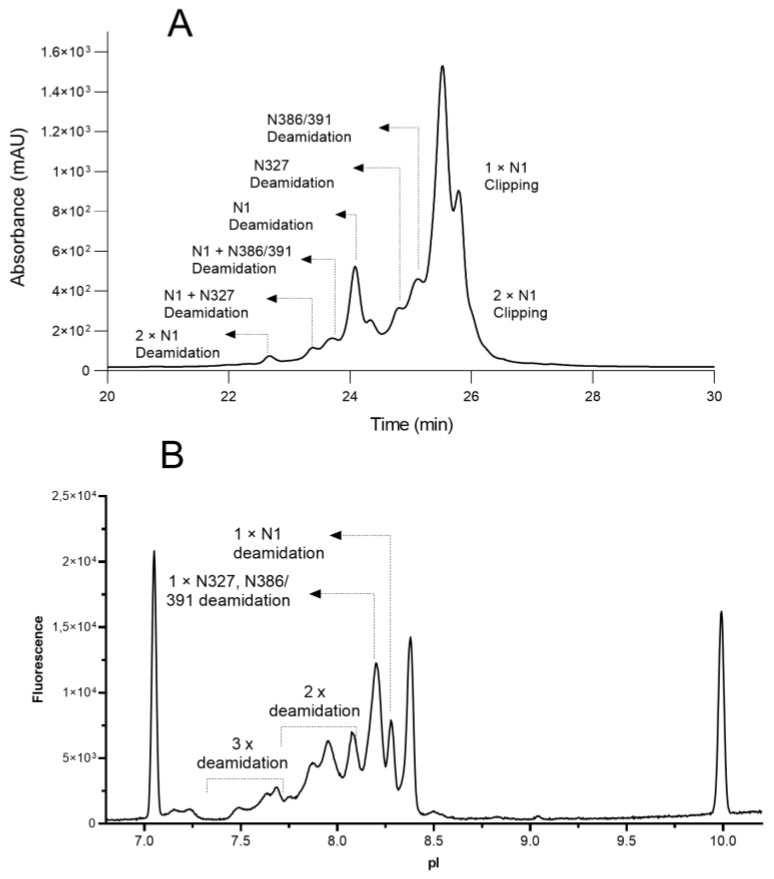
Assignments of peak identities for CEX chromatogram (**A**) and cIEF electropherogram (**B**).

**Figure 5 antibodies-12-00059-f005:**
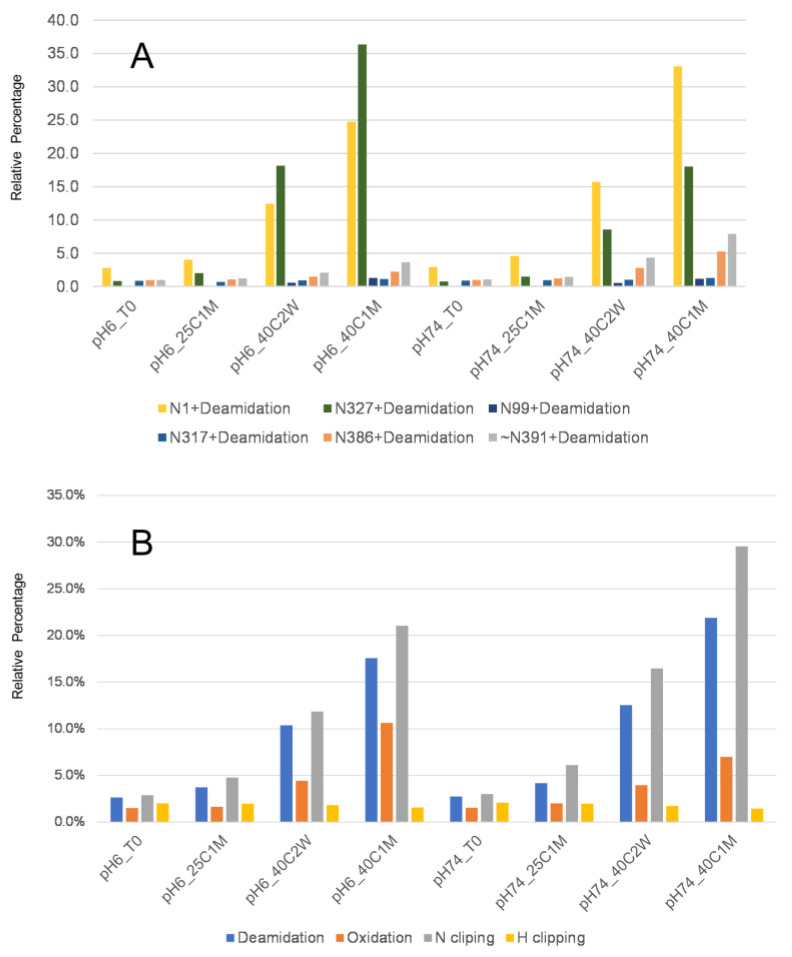
(**A**) Deamidation levels at different stress conditions. (**B**) N1 terminal peptide deamidation, oxidation and clipping under different stress conditions.

**Figure 6 antibodies-12-00059-f006:**
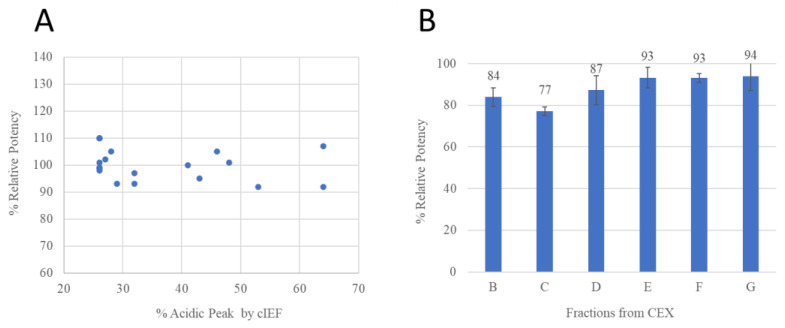
(**A**) A scatterplot of cIEF % acidic peaks and potency from the antibody stability studies; (**B**) The potency results of fractions from CEX. Error bars indicate standard deviations from triplicate runs.

**Figure 7 antibodies-12-00059-f007:**
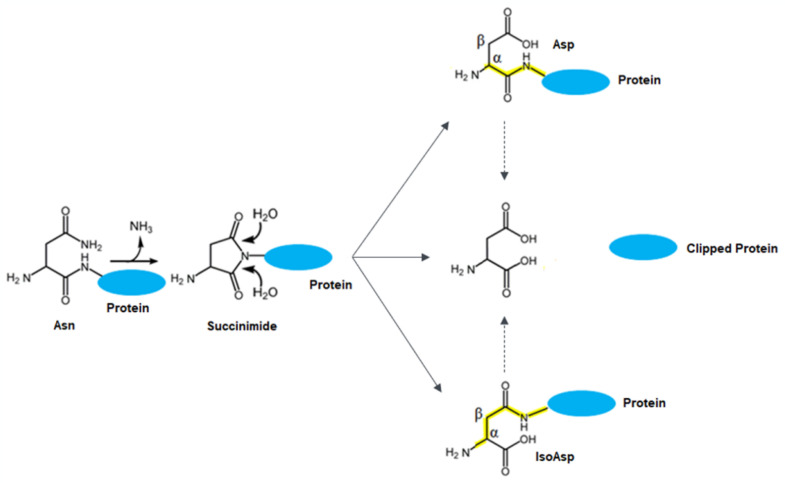
Proposed deamidation and clipping degradation pathway.

**Table 1 antibodies-12-00059-t001:** Charge variants percentage comparison between CEX and cIEF.

Sample Conditions	Acidic %	CEX Main %	Basic %	Acidic %	cIEF Main %	Basic
Control	19	78	2.5	29	66	4.4
25 °C, pH 6.0, 1 month	25	73	2.5	34	63	3.6
40 °C, pH 6.0, 2 weeks	50	34	16.1	59	37	3.8
40 °C, pH 6.0, 1 month	56	13	30.6	78	19.6	2.7

**Table 2 antibodies-12-00059-t002:** The major deamidations percentages and N-1 clipping in each CEX fraction.

Sample Conditions	Percentage of Deamidation			N-Terminal Clipping (%)
	N-1	N-327	N386/391	
A	78.2	8.5	5.0	1.4
B	44.4	2.6	4.7	2.9
C	4.6	30.3	7.4	5.2
D	1.4	11.3	11.5	8.5
E	0.4	0.8	5.8	5.8
F	0.5	0.7	4.9	27.1

**Table 3 antibodies-12-00059-t003:** N-terminal peptide variants identified.

Modification	Peptide	Theoretical *m*/*z* (z)	Observed *m*/*z*	R.T. (min)
Unmodified	NFMLTQPHSVSESPGK	879.9276 (+2)	879.9237	31.4
N1 deamidated	DFMLTQPHSVSESPGK	880.4196 (+2)	880.4162	32.3
	D(iso)FMLTQPHSVSESPGK	880.4196 (+2)	880.4163	33.0
N1 succinimide	SucFMLTQPHSVSESPGK	871.4143 (+2)	871.4152	40.0
M3 oxidation	NFM[O]LTQPHSVSESPGK	887.9251 (+2)	887.9222	28.7
N 1-clipped	FMLTQPHSVSESPGK	822.9061 (+2)	822.9057	30.3

## Data Availability

All related data and methods are presented in this paper the Appendix A. Additional inquiries should be addressed to the corresponding author.

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
