# Peer review of "Characterization of N-Terminal Asparagine Deamidation and Clipping of a Monoclonal Antibody"

_2073-4468, 2023, doi:10.3390/antib12030059_

Round 1

Reviewer 1 Report

The study focuses on the characterization of post-translational modifications (PTMs) in a therapeutic IgG molecule with N-terminal asparagine. Various PTMs, including deamidation and N-terminal pyroglutamate formation, are discussed, which can impact charge variants of monoclonal antibodies. The study uses chromatography (CEX and cIEF) and mass spectrometry for analysis, along with enzymatic digested peptide mapping. The research investigates the influence of temperature and incubation time on PTMs and characterizes a unique N-terminal asparagine clipping. The results show that the clipping is enriched under certain conditions and is likely related to the amyloidosis-linked gene. However, the impact on potency and aggregation of the IgG molecule is minimal.

The manuscript could benefit from improved section delineation and a more explicit connection of the findings to broader scientific implications.

Introduction: Commence with an introductory section underscoring the significance of investigating PTMs in biologics development, and highlight the relevance of the current study.

-Elaborate on the employed techniques—CEX, cIEF, mass spectrometry, and enzymatic peptide mapping—to elucidate the intricate processes used.

-In the Discussion section, discuss how these findings enhance our comprehension of PTMs and potential applications in biologics manufacturing.

- While the study effectively tackles its objectives, a more comprehensive exploration of the potential mechanisms underlying the observed N-terminal asparagine clipping would provide valuable insights.

- The study assumes that the N-terminal asparagine clipping does not significantly impact protein folding or aggregation. However, rigorous biophysical characterization, such as circular dichroism or differential scanning calorimetry, would provide a more complete assessment of structural integrity.

- The intriguing connection between N-terminal asparagine clipping and an amyloidosis-associated gene is proposed but not fully substantiated.

-Explore the intriguing prospect of the N-terminal asparagine clipping's involvement in amyloidosis and suggest avenues for future investigation.

- The investigation centers on a single therapeutic IgG molecule with N-terminal asparagine. Generalizing the findings to other antibody molecules or biologics with different sequences might require additional confirmatory studies.

-Address potential limitations or uncertainties in the findings, like the role of proteases in terminal cleavage and possible alternative mechanisms.

- The study primarily investigates short-term stability under accelerated and stressed conditions. Long-term stability over the entire shelf life of the therapeutic IgG molecule is not directly addressed.

-Conclude by highlighting the broader implications of the study's findings for biologics development, characterization, and quality control.

Author Response

Thank you very much for taking the time to review this manuscript. Please find the detailed responses attached and the corresponding revisions/corrections highlighted/in track changes in the re-submitted files. We made significant revision based on your suggestions. Please let us know if more need to be improved.

Reviewer 2 Report

This article discusses the significance of post-translational modifications (PTMs) in biologics, particularly monoclonal antibodies (mAbs). It highlights PTMs like deamidation and N-terminal pyroglutamate formation, explaining their effects on protein properties. The study focuses on an IgG molecule showing an unusual acidic peak increase, prompting a detailed analysis. The investigation reveals N-terminal asparagine deamidation products and a unique basic peak variant under heat stress. This research is the first to characterize these modifications' impact on mAb profiles, contributing to quality control in biologics manufacturing. The study demonstrates that N-terminal asparagine clipping in an IgG does not cause aggregation or affect its potency, confirming its structural stability. The findings offer insights into characterizing therapeutic antibodies with similar sequences in biologics process development. Here are two format points that need to be considerate:

1.      Please uniform all the graph labels to the up left corner and add error bars for the technical/ biological replicates.

2.      Please label the x-axis of Fig3A and Fig5.

Author Response

Thank you so much for reviewing this manuscript and provided valuable feedbacks. Please find the responses below and the corresponding revisions/corrections highlighted/in track changes in the re-submitted files. We made revisions on figures and some text refer to figures based on your suggestions. Please let us know if more need to be improved.

Point 1. Please uniform all the graph labels to the up left corner for the technical/ biological replicates.

Response: Except for the potency assay, all the other assays were not run in replicates because of the low variability of the assays and limited amount of the enriched samples. Figure 5A are from stability data which the %CV is < 20% as system suitability. Error bars have been added to Figure 6A based on triplicate runs.

Point 2.  Please label the x-axis of Fig3A and Fig5.

Response: Figures updated according to your suggestions.

Reviewer 3 Report

The authors present a manuscript with structure and a innovative approach in antibody characterization.

minor observations in spelling 

Author Response

Thank you for taking time to review this manuscript. We have carefully gone over the manuscript and corrected all the spelling. We also made some improvement. Please see the updated version with track-change on.

Round 2

Reviewer 1 Report

Thank you for addressing previous comments,